# GUIDED-TTS: TEXT-TO-SPEECH WITH UNTRANSCRIBED SPEECH

## ABSTRACT

Most neural text-to-speech (TTS) models require ⟨speech, transcript⟩ paired data from the desired speaker for high-quality speech synthesis, which limits the usage of large amounts of untranscribed data for training. In this work, we present Guided-TTS, a high-quality TTS model that learns to generate speech from untranscribed speech data. Guided-TTS combines an unconditional diffusion probabilistic model with a separately trained phoneme classifier for text-to-speech. By modeling the unconditional distribution for speech, our model can utilize the untranscribed data for training. For text-to-speech synthesis, we guide the generative process of the unconditional DDPM via phoneme classification to produce mel-spectrograms from the conditional distribution given transcript. We show that Guided-TTS achieves comparable performance with the existing methods without any transcript for LJSpeech. Our results further show that a single speaker-dependent phoneme classifier trained on multispeaker large-scale data can guide unconditional DDPMs for various speakers to perform TTS.

## 1 INTRODUCTION

Neural text-to-speech (TTS) models have been achieved to generate high-quality human-like speech given text (van den Oord et al. (2016); Shen et al. (2018)). In general, these TTS models are conditional generative models that encode text into the hidden representation and generate speech from the encoded representation. Early TTS models are autoregressive generative models which generate high-quality speech but suffer from slow synthesis speed due to the sequential sampling procedure (Shen et al. (2018); Li et al. (2019)). Owing to the development of non-autoregressive generative models, recent TTS models are capable of generating high-quality speech with faster inference speed (Ren et al. (2019); Ren et al. (2021); Kim et al. (2020); Popov et al. (2021)). Recently, high-quality end-to-end TTS models have been proposed that generate raw waveform from the text at once (Kim et al. (2021); Weiss et al. (2021); Chen et al. (2021b)).

Despite the high-quality and fast inference speed of speech synthesis, most TTS models can be trained only if the transcribed data of the desired speaker is given. While long-form untranscribed data, such as audiobooks or podcasts, is available on various websites, it is challenging to use these unpaired speech data to train existing TTS models. To utilize these untranscribed data, long-form untranscribed speech data has to be segmented into sentence-level, and then each segmented speech should be transcribed accurately. Since the existing TTS models must directly model the conditional distribution of speech given text, the direct usage of untranscribed data remains challenging to solve.

In this work, we propose Guided-TTS, an unconditional diffusion-based generative model trained on untranscribed data that leverages a phoneme classifier for text-to-speech synthesis. Trained on untranscribed speech data for the desired speaker, our unconditional diffusion probabilistic model learns to generate mel-spectrograms of the speaker without any context. As training data does not have to be aligned with text sequence for unconditional speech modeling, we simply use random chunks of untranscribed speech to train our unconditional generative model. This allows us to build training data without extra effort in modeling the speech of speakers for which only long-form untranscribed speech data is available.

To guide the unconditional DDPM for TTS, we train a framewise phoneme classifier on transcribed data and use the gradients of the classifier during sampling. Although our unconditional generative

model is trained without any transcript, Guided-TTS effectively generates mel-spectrograms given the transcript by guiding the generative process of unconditional DDPM with the phoneme classifier.

We demonstrate that the proposed method, TTS by guiding the unconditional DDPM, matches the performance of the existing conditional TTS models on LJSpeech. We further show that by training the phoneme classifier on multi-speaker paired dataset, Guided-TTS also shows comparable performance without seeing any transcript of LJSpeech, which shows the possibility to build a high-quality text-to-speech model without a transcript for the desired speaker. We encourage the readers to listen to samples of Guided-TTS trained on various untranscribed datasets on our demo page. [1]

## 2 BACKGROUND

### 2.1 DENOISING DIFFUSION PROBABLISTIC MODELS (DDPM) AND ITS VARIANT

DDPM (Ho et al. (2020)), which is recently proposed as a kind of probabilistic generative model, has been applied to various domains such as image (Dhariwal & Nichol (2021)) and audio (Chen et al. (2021a); Popov et al. (2021)). DDPM first defines a forward process that gradually corrupts data $X_0$ to a random noise $X_T$ across $T$ timesteps. The model learns the reverse process that follows the reverse trajectory of the predefined forward process to generate data from random noise.

Recently, there have been approaches to formulate the trajectory between data and noise as a continuous stochastic differential equation (SDE) instead of a discrete-time Markov process (Song et al. (2021b)). Grad-TTS (Popov et al. (2021)) introduces SDE formulation to TTS, which we have followed and used. According to the formulation of Grad-TTS, the forward process that corrupts data $X_0$ into the standard Gaussian noise $X_T$ is as follows:

$$dX_t = -\frac{1}{2} X_t \beta_t dt + \sqrt{\beta_t} dW_t, \tag{1}$$

where $\beta_t$ is a predefined noise schedule, $\beta_t = \beta_0 + (\beta_T - \beta_0)t$, and $W_t$ is a Wiener process. Anderson (1982) showed that the reverse process, which represents the trajectory from noise $X_T$ to $X_0$, can be formulated in SDE, which is defined as follows:

$$dX_t = (-\frac{1}{2} X_t - \nabla_{X_t} \log p_t(X_t))\beta_t dt + \sqrt{\beta_t} d\widetilde{W_t}, \tag{2}$$

where $\widetilde{W_t}$ is a reverse time Wiener process. Given the score, the gradient of log density with respect to data (*i.e.,* $\nabla_{X_t} \log p_t(X_t)$), for $t \in [0, T]$, we can sample data $X_0$ from random noise $X_T$ by solving Eq. (2). To generate data, the DDPM learns to estimate the score with the neural network $s_\theta$ parameterized by $\theta$.

To estimate the score, $X_t$ is sampled from the distribution derived from Eq. (1) given data $X_0$, which is as follows:

$$X_t|X_0 \sim \mathcal{N}(\rho(X_0, t), \lambda(t)), \tag{3}$$

where $\rho(X_0, t) = e^{-\frac{1}{2} \int_0^t \beta_s ds} X_0$, and $\lambda(t) = I - e^{-\int_0^t \beta_s ds}$. Then, the score can be derived from Eq. (3); $\nabla_{X_t} \log p_t(X_t|X_0) = -\lambda(t)^{-1}\epsilon_t$, given $X_0$ (Popov et al. (2021)). To train the model $s_\theta(X_t, t)$ for $\forall t \in [0, T]$, the following loss is used:

$$L(\theta) = \mathbb{E}_t \mathbb{E}_{X_0} \mathbb{E}_{\epsilon_t} \left[ \left\| s_\theta(X_t, t) + \lambda(t)^{-1}\epsilon_t \right\|_2^2 \right] \tag{4}$$

which is L2 loss as in previous works (Ho et al. (2020), Song et al. (2021b)).

Using the model $s_\theta(X_t, t)$, we can generate sample $X_0$ from noise by solving Eq. (2). Grad-TTS generates data $X_0$ from $X_T$ by setting $T = 1$ and using a fixed discretization strategy (Song et al. (2021b)):

$$X_{t-\frac{1}{N}} = X_t + \frac{\beta_t}{N}(\frac{1}{2}X_t + \nabla_{X_t} \log p_t(X_t)) + \sqrt{\frac{\beta_t}{N}} z_t \tag{5}$$

where $N$ is the number of steps to solve SDE, $t \in \{\frac{1}{N}, \frac{2}{N}, ..., 1\}$ and $z_t$ is a standard Gaussian noise.

---

[1] Demo : https://bit.ly/3oWhVJg

## 2.2 CLASSIFIER GUIDANCE

DDPM can be guided to generate samples with the desired condition without fine-tuning with the introduction of a classifier. Song et al. (2021b) used the unconditional DDPM to generate class-conditional images using a separately trained image classifier. Not only unconditional DDPM but also conditional DDPM can be guided using the classifier, which contributes to achieving state-of-the-art performance for class-conditional image generation (Dhariwal & Nichol (2021)).

For conditional generation, the classifier $p_t(y|X_t)$ is trained to classify the noisy data $X_t$ as the condition $y$. If Eq. (5) is modified, discretized SDE for the conditional generation can be obtained.

$$X_{t-\frac{1}{N}} = X_t + \frac{\beta_t}{N}(\frac{1}{2}X_t + \nabla_{X_t} \log p_t(X_t|y)) + \sqrt{\frac{\beta_t}{N}}z_t \tag{6}$$

$$\nabla_{X_t} \log p_t(X_t|y) = \nabla_{X_t} \log p_t(X_t) + \nabla_{X_t} \log p_t(y|X_t) \tag{7}$$

If the unconditional score and the classifier for the condition are given, the sample $X_0$ with the condition $y$ can be generated using Eq. (6).

## 3 GUIDED-TTS

In this section, we present Guided-TTS, which aims to build a high-quality text-to-speech model without the transcript of the target speaker. While other TTS models directly learn to generate speech from text, Guided-TTS models the unconditional distribution of speech to utilize speech-only data and guides the unconditional model to generate speech with a given text. To the best of our knowledge, Guided-TTS is the first TTS model that generates speech with the unconditional generative model.

Both unconditional speech modeling and controllable generation with the unconditional generative model are well known to be challenging. To tackle these challenges, we adopt a diffusion-based generative model for unconditional speech generation, which has the advantages of modeling complex distributions and easy controllability. Additionally, we introduce a phoneme classifier to guide the unconditional DDPM for TTS, which requires transcribed data for training.

In Guided-TTS, the generative model and the phoneme classifier are trained separately so that we can utilize different datasets to train each module. For instance, assume only the untranscribed speech data of the target speaker for TTS is available. With Guided-TTS, the phoneme classifier can still be trained on other transcribed datasets that contain rich, large-scale multi-speaker data. The phoneme classifier trained in such a manner effectively guides the generative model, trained only using the untranscribed speech data of the target speaker. By doing so, with the two modules independent of each other, we can achieve the text-to-speech model without any transcript of the target speaker.

Guided-TTS consists of 4 modules: the unconditional DDPM, the phoneme classifier, the duration predictor, and the speaker encoder, as shown in Fig. 1. Unconditional DDPM is a module that learns to generate mel-spectrogram unconditionally, and the remaining three modules are for TTS synthesis by guidance. We will explain the unconditional DDPM in Section 3.1, followed by the method of guiding the unconditional model for TTS in Section 3.2.

### 3.1 UNCONDITIONAL DDPM

Our unconditional DDPM models the unconditional distribution of speech $P_X$ without any transcript. We assume that the training data for the diffusion-based model has tens of hours of untranscribed speech data from the target speaker $S$ for TTS. As our generative model learns only with speech data, training samples do not need to be aligned with the text. Thus, we use random chunks of untranscribed speech as training data to reduce the burden of not only speech transcription but also segmentation when only the long-form speech data is available for the target speaker.

Given a mel-spectrogram $X = X_0$, we define a forward process as in Eq. (1), which gradually corrupts data into noise, and approximate the reverse process in Eq. (2) by estimating the unconditional score $\nabla_{X_t} \log p(X_t)$ for each timestep $t$. At each iteration, $X_t, t \in [0, 1]$ is sampled from the mel-spectrogram $X_0$ as in Eq. (3), and score is estimated with the neural network $s_\theta(X_t, t)$ parameterized by $\theta$. The training objective of our unconditional model is in Eq. (4).

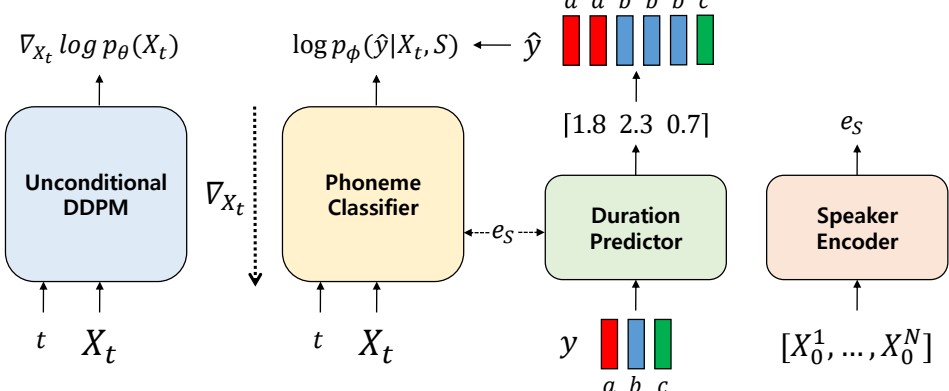

Figure 1: The overall architecture of Guided-TTS. The unconditional DDPM learns to generate speech $X_0$ without the transcript. The other modules, the phoneme classifier, duration predictor, and speaker encoder are for guiding the unconditional DDPM to generate conditional samples given $y$. The speaker embedding $e_S$ is only used for training speaker-dependent modules.

Similar to Grad-TTS (Popov et al. (2021)), we regard mel-spectrogram as a 2D image with a single channel and use the U-Net architecture (Ronneberger et al. (2015)) as $s_\theta$. We use the same size of the architecture used to model $32 \times 32$ sized images in Ho et al. (2020) to capture long-term dependencies without any text information, while Grad-TTS uses smaller architecture for the conditional distribution modeling.

## 3.2 TEXT-TO-SPEECH VIA CLASSIFIER GUIDANCE

For TTS synthesis, we use a framewise phoneme classifier to guide the unconditional DDPM. As shown in Fig. 1, in order to generate mel-spectrogram given text, our duration predictor outputs the duration for each text token and expands the transcript $y$ to frame-level phoneme label $\hat{y}$. Then, we sample a random noise $X_T$ of the same length as $\hat{y}$ from the standard normal distribution, and we can generate conditional samples by replacing the unconditional score in Eq. (5) with the conditional score. As in Eq. (8), we can estimate the conditional score on the left side by adding the two terms on the right side: the first term is obtained from the unconditional DDPM, and the second term can be computed with the phoneme classifier. That is, we achieve to build a text-to-speech model with the unconditional generative model for the speech by adding the gradient of the phoneme classifier during the generative process.

$$\nabla_{X_t} \log p(X_t|\hat{y}, spk = S) = \nabla_{X_t} \log p_\theta(X_t|spk = S) + \nabla_{X_t} \log p_\phi(\hat{y}|X_t, spk = S) \quad (8)$$

If the transcribed speech dataset of the target speaker $S$ is available, we train all the unconditional DDPM, the phoneme classifier, and the duration predictor on the dataset of the speaker $S$ for TTS. Otherwise, if we only have the untranscribed speech data for the target speaker $S$, we first train unconditional DDPM on the untranscribed speech data and leverage a large-scale multi-speaker transcribed speech dataset to train the phoneme classifier and the duration predictor. In this case, to guide unconditional DDPM for the target speaker $S$, our phoneme classifier and duration predictor are designed to be speaker-dependent modules and to generalize well to the unseen speaker $S$ during training. We provide the speaker embedding extracted from the pre-trained speaker verification network as condition to both modules, as described in Fig. 1. We describe each module required for guidance below.

**Phoneme Classifier** The phoneme classifier is a network trained on transcribed data that recognizes the phoneme corresponding to each frame of the input mel-spectrogram. To train the framewise phoneme classifier, we align transcript and speech using a forced alignment tool, Montreal Forced Aligner (MFA) (McAuliffe et al. (2017)), and extract the frame-level phoneme label $\hat{y}$. The phoneme classifier is trained to classify the corrupted mel-spectrogram $X_t$ sampled from Eq. (3) as the frame-level phoneme label $\hat{y}$. The training objective of the phoneme classifier is to maximize the

expectation of cross-entropy between the phoneme label $\hat{y}$ and the output probability with respect to $t \in [0, 1]$.

We use a WaveNet-like architecture (van den Oord et al. (2016)) as a phoneme classifier, and time embedding $e_t$, which is extracted in the same way as in Popov et al. (2021), is used as a global condition in WaveNet to provide information about the noise level of the corrupted input $X_t$ at timestep $t$. Additionally, for speaker-dependent classification, we use the speaker embedding $e_S$ from the speaker encoder as the global condition.

**Duration Predictor** Duration predictor is a module that predicts the duration of each text token for a given text sequence $y$. We extract the duration label of each text token using MFA for the same data on which the phoneme classifier is trained. The duration predictor is trained to minimize L2 loss between the duration label and the estimated duration in the log-domain, and we round up the estimated duration during inference. The architecture of the duration predictor is the same as that of Glow-TTS (Kim et al. (2020)) with the text encoder. We concatenate the text embedding and the speaker embedding $e_S$ for training a speaker-dependent module.

**Speaker Encoder** Speaker encoder encodes the speaker information from the input mel-spectrogram and outputs the speaker embedding $e_S$. Similar to Jia et al. (2018), we train a speaker encoder with GE2E loss (Wan et al. (2018)) on the speaker verification dataset and use speaker encoder to condition speaker-dependent modules. To train speaker-dependent modules, we use speaker embedding $e_S$ extracted from the clean mel-spectrogram $X_0$ for each training data. For guidance, we average and normalize the speaker embeddings of untranscribed speech for the desired speaker $S$ to extract $e_S$.

### 3.2.1 NORM-BASED GUIDANCE

---
**Algorithm 1** Norm-based Guidance

---
$\hat{y}$: framewise phoneme label, $s$: gradient scale, $\tau$: temperature
$\theta$: parameter of unconditional DDPM:, $\phi$: parameter of phoneme classifier
$X_1 \sim \mathcal{N}(0, \tau^{-1}I)$
**for** $i = N$ **to** $1$ **do**
    $t \leftarrow \frac{i}{N}$
    $\alpha_t \leftarrow \|\nabla_{X_t} \log p_\theta(X_t)\| / \|\nabla_{X_t} \log p_\phi(\hat{y}|X_t)\|$
    $z_t \sim \mathcal{N}(0, \tau^{-1}I)$
    $X_{t-\frac{1}{N}} \leftarrow X_t + \frac{\beta_t}{N}(\frac{1}{2}X_t + \nabla_{X_t} \log p_\theta(X_t) + s \cdot \alpha_t \nabla_{X_t} \log p_\phi(\hat{y}|X_t))) + \sqrt{\frac{\beta_t}{N}}z_t$
**end for**
**return** $X_0$

---

Initially, we scaled classifier gradient $\nabla_{X_t} \log p_\phi(\hat{y}|X_t, spk = S)$ in Eq. (8) using gradient scale $s$, which is the scaled version of Song et al. (2021b). However, when guiding the unconditional DDPM with the framewise phoneme classifier, we found that the norm of the unconditional score suddenly increases near the data. That is, the closer to the data, the phoneme classifier has little effect on the generative process of DDPM. The norm of unconditional score $\|\nabla_{X_t} \log p_\theta(X_t)\|$ according to timestep $t$ is shown in Fig. 4 of Appendix A.4. Here, we propose norm-based guidance to guide the unconditional DDPM better in terms of generating speech conditioned on frame-level phoneme label $\hat{y}$. Norm-based guidance is a method of scaling the norm of the classifier gradient in proportion to the norm of the score in order to prevent the effect of the gradient from being insignificant as the score rises steeply. The ratio between the norm of the scaled gradient and the norm of the score is defined as the gradient scale $s$. By adjusting $s$, we can determine how much the classifier gradient contributes to the guidance of unconditional DDPM. We also introduced the temperature parameter $\tau$ when guiding the DDPM. We observed that tuning $\tau$ to a value greater than 1 helps generate high-quality mel-spectrograms.

## 4 EXPERIMENTS

**Datasets** For experiments using a transcribed single dataset, we train our unconditional DDPM and phoneme classifier respectively on speech-only LJSpeech (Ito (2017)) and speech-transcript paired

LJSpeech to match the settings of baseline TTS models. In addition, since our method makes use of a set of separately trained unconditional DDPM and phoneme classifier, they can also be trained with different datasets respectively. Here, by training the unconditional DDPM with a speech-only target speaker dataset and the phoneme classifier with a transcribed large-scale multi-speaker dataset, we can build a high-quality TTS model from any specific speaker dataset without any transcript. For the corresponding experiments, we use only the speech data from LJSpeech, Hi-Fi TTS (Bakhturina et al. (2021)), and Blizzard 2013 (King & Karaiskos (2013)) for training unconditional DDPMs, and LibriTTS (Zen et al. (2019)), a large-scale multi-speaker dataset with approximately 585 hours of speech uttered by 2456 speakers with corresponding texts, for training the speaker-dependent phoneme classifier to guide the various unconditional generative models. To extract the speaker embedding $e_S$ from each utterance, we train a speaker encoder on VoxCeleb2 (Chung et al. (2018)), which is a speaker verification dataset that contains more than 1M utterances for 6112 speakers.

LJSpeech is a 24-hour single female speaker dataset consisting of 13,100 audio clips. Dataset is randomly split; 12,500 samples for the training set, 100 samples for the validation set, and 500 samples for the test set. Hi-Fi TTS is a multi-speaker dataset with 6 females and 4 males, and each speaker's data consists of at least 17 hours of speech. We select three of them (two males and one female) and use them for training three unconditional DDPMs, respectively. Blizzard 2013 is an 147 hours segmented and unsegmented audiobook data read by a single female speaker. We only use 5-second randomly clipped audio of unsegmented data to show that our model does not require text labeling or aligning for untranscribed data. Only audio files are used to train unconditional DDPM.

**Training Details** We convert text into International Phonetic Alphabet (IPA) phoneme sequences using open-source software (Bernard (2021)). To extract the mel-spectrogram, we use the same hyperparameters as Glow-TTS (Kim et al. (2020)). All modules are trained using Adam optimizer with a learning rate of $0.0001$. For the unconditional model and the phoneme classifier, $\beta_0 = 0.05, \beta_1 = 20$ are used for beta schedule. For unconditional model, the base model has the same architecture and hyperparameters as DDPM (Ho et al. (2020)) used for $32 \times 32$ sized image modeling, and for ablation, we use the same architecture and hyperparameters for the small model as Grad-TTS (Popov et al. (2021)). Other details and hyperparameters are described in the Appendix A.2.

**Evaluation** To compare the performance of models, we use pre-trained models and the official implementations of Glow-TTS and Grad-TTS.[2] For Glow-TTS, we use a pre-trained model with blank tokens between phonemes and use $\tau = 1.5$. We use the same hyperparameters as the official implementation, $\tau = 1.5$, and the number of reverse steps $N = 50$ for Grad-TTS. For our model, we set $\tau = 1.5$, gradient scale $s = 0.3$, and the number of reverse steps $N = 50$. We use official implementation and pre-trained models of HiFi-GAN as a vocoder.[3]

To show whether Guided-TTS with the norm-based guidance generates the sentences of the given text accurately, we measure the character error rate (CER) and word error rate (WER) for each model, which are metrics commonly used in automatic speech recognition (ASR). We use the pre-trained CTC-based conformer (Graves et al. (2006), Gulati et al. (2020)) model provided by NEMO toolkit (Kuchaiev et al. (2019)) to compute the metrics.

## 5 RESULTS

### 5.1 MODEL COMPARISON

We compare the performance of audio samples by measuring the 5-scale mean opinion score (MOS) on LJSpeech using Amazon Mechanical Turk. In addition, through CER and WER, we check whether the generated sample of Guided-TTS faithfully reflects the text. To calculate CER and WER, we first synthesize the speech of a given text for each model and provide it to the ASR model to extract the text corresponding to the generated sample. We then measure the CER and WER between the ground truth text and the text obtained from the ASR model. For evaluation, we randomly select 50 samples drawn from the test set of LJSpeech, and the results are shown in Table 1.[4] In Table 1, Guided-TTS-T (T:Transcribed) is specified for the case of training the unconditional

---

[2]Glow-TTS: https://bit.ly/3kS315K, Grad-TTS: https://bit.ly/3qTCmcJ

[3]HiFi-GAN: https://bit.ly/3FxBv5x

[4]Demo : https://bit.ly/3oWhVJg

DDPM and the other modules on the speech data of the target speaker. The quality of Guided-TTS-T on LJSpeech shows that our model achieves TTS performance comparative to the baseline models. Furthermore, the ASR metrics of Guided-TTS-T indicate that our model generates speech samples according to given text stably like the existing TTS models do. Unlike Grad-TTS, which leverages the conditional DDPM for TTS, our unconditional DDPM requires a large model size for modeling long-term dependency of speech without text. The qualitative results of Guided-TTS-T with the small unconditional model (Guided-TTS-T (small)), whose architecture is the same as the decoder of Grad-TTS, demonstrate the importance of modeling capability of unconditional DDPM for the naturalness of the samples and accurate pronunciation.

Guided-TTS-U (U:Untranscribed) is the case that we train unconditional DDPM using only the target speaker's text-free speech and guide it using a speaker-dependent phoneme classifier separately trained on LibriTTS. Guided-TTS-U shows performance and ASR metrics comparable to other TTS models, which require transcribed data of the desired speaker for training. This demonstrates that Guided-TTS enables building a high-quality TTS model using the untranscribed speech of the target speaker. Samples of all models are available on the demo page.

Table 1: Mean Opinion Score (MOS) with 95% confidence intervals and Automated Speech Recognition (ASR) metrics of TTS models for LJSpeech. The ASR metric indicates character error rate (CER) and word error rate (WER) measured by the pre-trained ASR model. The unconditional DDPM in Guided-TTS uses only untranscribed speech of LJSpeech for training. **Guided-TTS-T** is a model whose phoneme classifier and duration predictor are trained using the transcript of LJSpeech. **Guided-TTS-U** uses a multi-speaker dataset instead of the transcript of LJSpeech to train phoneme classifier and duration predictor.

| Method | Transcript of LJSpeech | 5-scale MOS | CER/WER(%) |
|---|---|---|---|
| Ground Truth | | $4.46 \pm 0.05$ | 0.79 / 3.77 |
| Ground Truth (Mel + HiFi-GAN) | | $4.25 \pm 0.07$ | 1.05 / 4.08 |
| Glow-TTS | ✓ | $4.10 \pm 0.10$ | 1.09 / 5.03 |
| Grad-TTS | ✓ | $4.25 \pm 0.09$ | 1.31 / 5.55 |
| Guided-TTS-T | ✓ | $4.18 \pm 0.07$ | 1.20 / 4.71 |
| Guided-TTS-T (small) | ✓ | $4.01 \pm 0.07$ | 1.65 / 5.97 |
| Guided-TTS-U | ✗ | $4.18 \pm 0.07$ | 1.26 / 4.61 |

Table 2: Mean Opinion Score (MOS) of Guided-TTS-U with various untranscribed datasets with 95% confidence intervals. In Guided-TTS-U, the phoneme classifier and the duration predictor are trained on LibriTTS dataset, and the speaker encoder is trained on VoxCeleb2 dataset.

| Method | Untranscribed Data | 5-scale MOS | Total duration (hrs) | Gender |
|---|---|---|---|---|
| | LJSpeech | $4.18 \pm 0.07$ | 24 | Female |
| | Hi-Fi TTS ID: 92 | $3.97 \pm 0.08$ | 27.3 | Female |
| Guided-TTS-U | Hi-Fi TTS ID: 6097 | $3.82 \pm 0.08$ | 30.1 | Male |
| | Hi-Fi TTS ID: 9017 | $3.72 \pm 0.08$ | 58.0 | Male |
| | Blizzard 2013 | $3.85 \pm 0.08$ | 149.4 | Female |

## 5.2 GENERALIZATION TO DIVERSE DATASETS

We show that our model can synthesize high-quality speech of LJSpeech using untranscribed speech in the previous section. Since our model trains the unconditional model and the classifier separately, TTS for various untranscribed datasets is possible with a single classifier trained with a large-scale paired multi-speaker dataset. The performance of our multiple TTS models using the untranscribed speech of target speaker is presented in Table 2. For evaluation, we used the same 50 randomly chosen sentences from the test set of LJSpeech. In Table 2, we show that if we have the phoneme classifier, duration predictor, and speaker encoder trained on a large-scale multi-speaker dataset, we can

build a TTS model using untranscribed speech data of any speaker without additional burdens such as transcription and segmentation. In particular, Guided-TTS-U synthesizes high-quality samples even when trained on a randomly cropped unsegmented Blizzard 2013 dataset. Our method enables TTS for diverse untranscribed datasets with various characteristics (*e.g.,* gender, American/British accent, prosody). There are samples on the demo page for diverse datasets.

## 5.3 ANALYSIS ON NORM-BASED GUIDANCE

We also compare the proposed norm-based classifier guidance with the classifier guidance used in previous works (Song et al. (2021b); Dhariwal & Nichol (2021)). A model that performs the conditional generation task with classifier guidance sometimes generates samples of conditions other than the target condition (Song et al. (2021b)). Similarly, we observe that Guided-TTS with the previous guidance method produces mispronounced samples given text. To show the effect of norm-based guidance and adjustment of gradient scale, we measure CER of Guided-TTS-T and Guided-TTS-U for LJSpeech according to the gradient scale $s$. We explore the gradient scale $s$ within $[0.5, 1.0, ..., 4.5]$ for the scaled version of the previous method (Song et al. (2021b)), and $[0.1, 0.2, ..., 1.0]$ for the norm-based guidance.

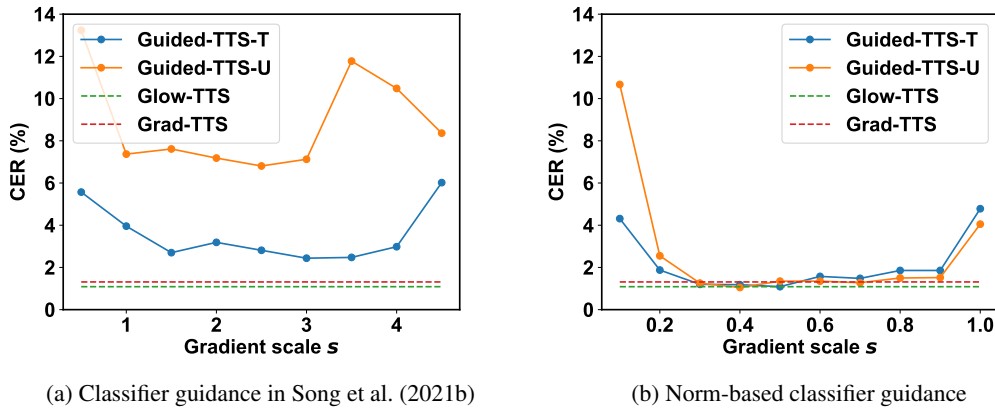

(a) Classifier guidance in Song et al. (2021b)         (b) Norm-based classifier guidance

Figure 2: CER according to gradient scales, (a) CER of Guided-TTS with the classifier guidance (scaled version of Song et al. (2021b)), (b) CER of Guided-TTS with the norm-based guidance

Fig. 2 presents the CER of the Guided-TTS with the classifier guidance in the previous works and the proposed norm-based guidance. As shown in Fig. 2a, the sample generated by the existing guidance method shows a far worse CER than the existing TTS models, which indicates that it is not suitable for TTS. On the other hand, Fig. 2b shows that the proposed guidance method with the appropriate gradient scale helps generate samples given text sentences accurately like the existing TTS models.

If the gradient scale is too small, the effect of the classifier gradient is negligible, and generated samples do not reflect the given text. On the other hand, we observed that guidance with a gradient scale too large deteriorates the sample quality. For the proposed norm-based guidance, we set the default gradient scale $s$ to $0.3$, which generates high-quality samples that exactly match the given text. Samples for multiple gradient scales with each guidance method are on the demo page.

## 6 RELATED WORK

**Unconditional Speech Generation** In general, the unconditional speech generative model (van den Oord et al. (2016); Vasquez & Lewis (2019)), which models audio without any information, is more challenging than the conditional generative model that synthesizes speech using text or mel-spectrograms. Several works attempt to unconditionally generate raw waveforms (van den Oord et al. (2016); Donahue et al. (2019)) or to model the unconditional distribution of latent code or mel-spectrogram of audio (van den Oord et al. (2017); Vasquez & Lewis (2019)) instead of directly modeling raw waveforms. Most existing unconditional models have only been used for unconditional audio modeling and no other purpose. To the best of our knowledge, this is the first applica-

tion of an unconditional model for TTS with appropriate guidance to enable speech synthesis using untranscribed data of the target speaker.

**Text-to-Speech Models** Most text-to-speech (TTS) models are composed of two parts: a model that generates intermediate features (*e.g.,* mel-spectrogram) from text (Shen et al. (2018)) and the vocoder, which synthesizes raw waveforms from intermediate features (van den Oord et al. (2016)). The autoregressive model is used for the text-to-intermediate feature model (Wang et al. (2017); Shen et al. (2018); Ping et al. (2018); Li et al. (2019)) and vocoder (van den Oord et al. (2016); Kalchbrenner et al. (2018)) to perform high-quality TTS. Since the autoregressive models generate samples in a sequential manner, their inference speed is slow. Thus, various parallel TTS models have been proposed to improve the sampling speed. Flow-based generative models (Kingma & Dhariwal (2018)) and feed-forward models have been proposed for text-to-mel-spectrogram models (Ren et al. (2019); Ren et al. (2021); Kim et al. (2020); Shih et al. (2021)) and vocoders (Oord et al. (2018); Prenger et al. (2019); Kim et al. (2019)). Also, variational autoencoder (Kingma & Welling (2014)) based models (Lee et al. (2020); Liu et al. (2021)), diffusion (Ho et al. (2020)) based models (Chen et al. (2021a); Kong et al. (2021); Popov et al. (2021); Jeong et al. (2021)), GAN (Goodfellow et al. (2014)) based models (Kumar et al. (2019); Bińkowski et al. (2019); Kong et al. (2020)) have been proposed as high-quality speech synthesis models with parallel sampling schemes.

Recently, end-to-end TTS models have been proposed, such as Ren et al. (2021), Donahue et al. (2021), Weiss et al. (2021), Kim et al. (2021), and Chen et al. (2021b). Most previous TTS models perform conditional generation tasks using transcribed speech data of the target speaker. We propose a new TTS model that learns to generate speech of the target speaker using untranscribed data and guides it through a phoneme classifier trained with other transcribed datasets. By guiding the unconditional model with the independently trained classifier, we achieve a high-performance TTS model using untranscribed data of the desired speaker without sentence-level segmentation or transcription.

**Diffusion-based Generative Models** DDPM (Ho et al. (2020)) has undergone several theoretical developments (Song et al. (2021b)) and produces high quality samples in many domains (Ho et al. (2020); Dhariwal & Nichol (2021); Chen et al. (2021a); Popov et al. (2021); Luo & Hu (2021)). Many theoretical and practical breakthroughs (Nichol & Dhariwal (2021); Lam et al. (2021); Chen et al. (2021a); Song et al. (2021a); Watson et al. (2021); Kong & Ping (2021); Jolicoeur-Martineau et al. (2021)) have been proposed, and a continuous version of DDPM, an SDE-based model (Song et al. (2021b); Popov et al. (2021)) is also presented. DDPM has also shown strong performance in speech synthesis (Chen et al. (2021a); Kong et al. (2021); Popov et al. (2021); Jeong et al. (2021)).

A pre-trained unconditional DDPM can be used for various tasks such as imputation (Song et al. (2021b)), and controllable generation (Song et al. (2021b)). In particular, the controllable generation allows Dhariwal & Nichol (2021) to achieve state-of-the-art performance in class-conditional image generation by guiding the DDPM using a gradient from the classifier trained on the same dataset as DDPM. We introduce the classifier guidance method of unconditional DDPM to text-to-speech synthesis. Our unconditional DDPM and the phoneme classifier can be trained using different datasets, making it possible to build a TTS model with the target speaker's untranscribed speech.

## 7 CONCLUSION

In this work, we present Guided-TTS, a new type of TTS model that generates speech given transcript by guiding the unconditional diffusion-based model for speech. As Guided-TTS models unconditional distribution for speech, the proposed model can construct a TTS model using the desired speaker's untranscribed data. Thanks to the properties of diffusion-based generative models, our unconditional generative model can generate a speech when a transcript is given by introducing the separately trained phoneme classifier. To the best of our knowledge, Guided-TTS is the first TTS model to leverage the unconditional generative model for speech. We showed that Guided-TTS matches the performance of the previous TTS models on LJSpeech. Moreover, Guided-TTS without the transcript of the target speaker generates samples comparable to existing models using transcripts by training its phoneme classifier on a large-scale multi-speaker dataset. We also showed that the single well-performed phoneme classifier can guide various unconditional DDPMs to generate high-quality sample. We believe that Guided-TTS can reduce the burden of constructing training datasets for high-quality TTS.

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

# A  APPENDIX

## A.1  INPAINTING

We perform the inpainting task to show how well the unconditional DDPM learns the dependencies in mel-spectrogram. The pre-trained unconditional DDPM fills out the masked part of the mel-spectrogram. We use samples from three speakers; one female speaker (ID: 92) and one male speaker (ID: 6097) from Hi-Fi TTS and a female speaker from LJSpeech. Two cross-shaped masks (LJSpeech, Hi-Fi TTS male) and one binarized MNIST (LeCun & Cortes (2010)) mask (Hi-Fi TTS female) are used for masking. We set 1000 as the number of reverse steps $N$ and $\tau = 1.5$ for inpainting. The method of inpainting is the same as Song et al. (2021b), and the algorithm is as follows:

---

**Algorithm 2** Inpainting Mel-spectrogram

---

Binary Mask: $M$, Original mel-spectrogram: $\hat{X}_0$
$\theta$: parameter of unconditional DDPM
$X_1 \sim \mathcal{N}(0, \tau^{-1}I)$
**for** $i = N$ **to** $1$ **do**
$\quad t \leftarrow \frac{i}{N}$
$\quad \rho(\hat{X}_0, t) \leftarrow e^{-\frac{1}{2}\int_0^t \beta_s ds}\hat{X}_0$
$\quad \lambda(t) \leftarrow I - e^{-\int_0^t \beta_s ds}$
$\quad \hat{X}_t \sim \mathcal{N}(\rho(\hat{X}_0, t), \lambda(t))$
$\quad X_t \leftarrow X_t \odot M + \hat{X}_t \odot (1 - M)$
$\quad z_t \sim \mathcal{N}(0, \tau^{-1}I)$
$\quad X_{t-\frac{1}{N}} \leftarrow X_t + \frac{\beta_t}{N}(\frac{1}{2}X_t + \nabla_{X_t}\log p_\theta(X_t)) + \sqrt{\frac{\beta_t}{N}}z_t$
**end for**
**return** $X_0 \odot M + \hat{X}_0 \odot (1 - M)$

---

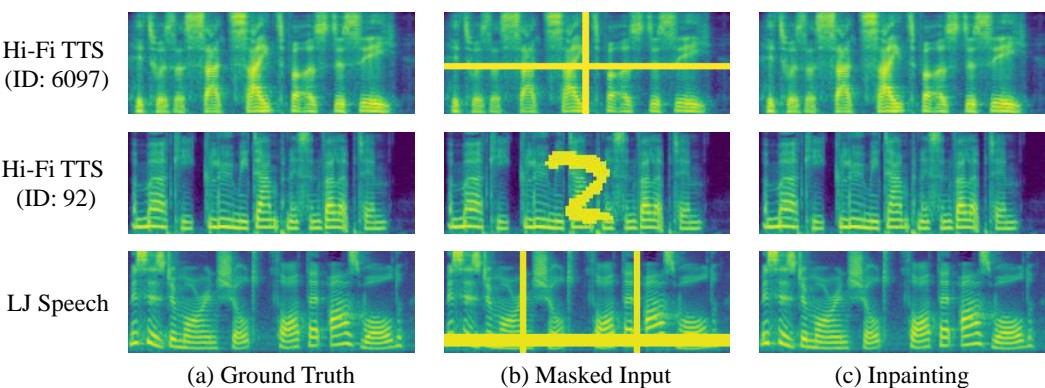

|  | (a) Ground Truth | (b) Masked Input | (c) Inpainting |

Figure 3: Mel-spectrogram inpainting results of unconditional DDPM trained on LJSpeech, and two speakers (Speaker ID: 92, 6097) from Hi-Fi TTS.

The inpainting results are shown in Fig. 3, where (a) is the original mel-spectrogram, (b) is the masked mel-spectrogram, and (c) is the result of inpainting on the masked part. As shown in Fig. 3, we show that our unconditional DDPM learns the adjacent frequency and temporal dependencies of the mel-spectrogram. Samples of inpainting results are provided on the demo page.

## A.2  TRAINING DETAILS AND HYPERPARAMTERS

In this section, we cover the training details and detailed hyperparameters. In Guided-TTS-T, all modules are trained on LJSpeech. In Guided-TTS-U, we use only untranscribed data of the various target speakers for training unconditional DDPMs and train a phoneme classifier and a duration

predictor on LibriTTS. Alignment labels are required for training the phoneme classifier and the duration predictor, and we train MFA to extract the alignment using the same dataset used to train the phoneme classifier and duration predictor.

For both Guided-TTS-T and Guided-TTS-U, the unconditional models are trained with batch size 16 for all datasets. The duration predictor is trained for 20 epochs with batch size 64. The phoneme classifier of Guided-TTS-T uses a WaveNet-like structure with 256 residual channels, 6 residual blocks stacks of 3 dilated convolution layers, and is trained for 1000 epochs with batch size 64. For Guided-TTS-U, we use a WaveNet-like structure with 512 residual channels, 3 residual blocks stacks of 6 dilated convolution layers for the phoneme classifier, and it is trained for 140 epochs with batch size 64. The speaker encoder is a two-layer LSTM with 768 channels followed by a linear projection layer to extract 256-dimensional speaker embedding $e_S$, and trained for 300K iterations.

For sampling, we use the last checkpoint for the unconditional DDPM and the speaker encoder. We use the checkpoint of the epoch with the best metric for the phoneme classifier (validation accuracy) and the duration predictor (validation loss).

## A.3 Hardware and Sampling Speed

We conduct all experiments and evaluations using NVIDIA's RTX 8000 with 48GB memory. Although our model is not focused on fast inference, Guided-TTS can perform real-time speech synthesis on GPU for $N = 50$, which is the number of reverse steps we use for evaluation. We measure the sampling speed of Guided-TTS using a real-time factor (RTF). We also measure how much time it takes to compute the unconditional score ($\nabla_{X_t} \log p_\theta(X_t)$) and gradient of the classifier ($\nabla_{X_t} \log p_\phi(\hat{y}|X_t)$). Guided-TTS-T achieves an RTF of 0.56, of which 0.39 is used to calculate the score and 0.16 is used for classifier gradient calculation. Similarly, Guided-TTS-U achieves an RTF of 0.63, 0.39 for computing the score and 0.23 for computing the gradient of the classifier.

## A.4 Norm of the Unconditional Score and Classifier Gradient

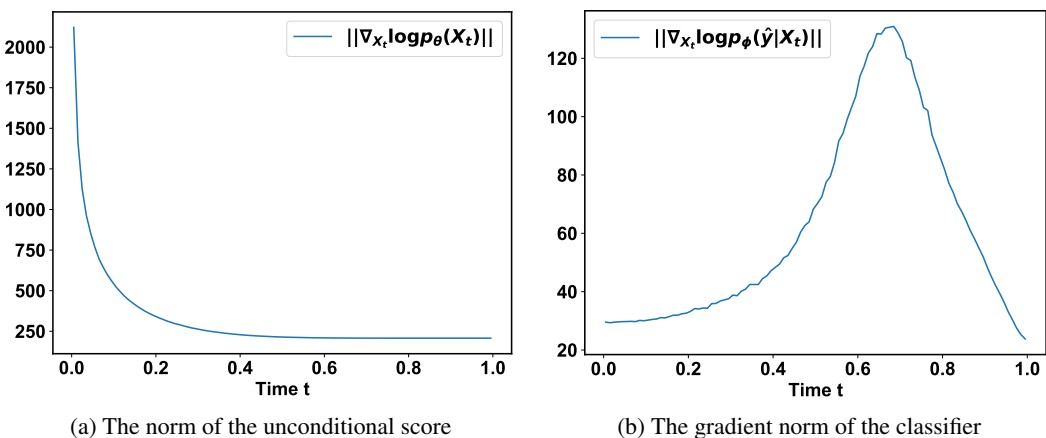

(a) The norm of the unconditional score      (b) The gradient norm of the classifier

Figure 4: The norm of the unconditional score and the classifier gradient for each timestep

The norm of the unconditional score and the gradient norm of the classifier for each timestep are shown in Fig. 4. We sample $X_t$ at a total of 1000 timesteps ($t \in (\frac{1}{2000}, \frac{3}{2000}, ..., \frac{1999}{2000})$) using the Eq. (3) for all 500 samples from the test set of the LJSpeech. We then obtain the norm of the unconditional score and the gradient of the classifier using the sampled $X_t$ with the modules of Guided-TTS-T. Each norm is averaged over 500 samples for each timestep. As shown in Fig. 4a, the norm of the unconditional score rises steeply around $t = 0$. This is about 70 times larger than the norm of the classifier gradient near $t = 0$ (Fig. 4b), which significantly reduces the effect of the classifier guidance. To alleviate this problem, we propose the norm-based guidance in Section 3.2.1, which helps prevent both the gradient of the classifier from being ignored and the issue of synthesized speech not matching the text.

