# OpenReview forum: "Guided-TTS:Text-to-Speech with Untranscribed Speech"
_ICLR.cc/2022/Conference — ICLR 2022 Submitted_

### Official Review · Reviewer_6P3J · 2021-11-01

**Correctness:** 3
**Technical Novelty And Significance:** 2
**Empirical Novelty And Significance:** 1
**Recommendation:** 3
**Confidence:** 4

**Main Review:**

Strengths:

- The application of the method from Song et al to TTS is novel.
- The demonstrated method allows the use of lots of untranscribed data for TTS, which is very useful.

Weaknesses:

- The novelty is limited. It is just a direct application of a previously proposed method (Song et al) on TTS.
- The experiments and analysis are weak.
- Even with the use of more data, the method is not able to outperform existing models.
- No published code.

Unconditional DDPM can be trained unsupervisedly, yes.
But not the phoneme classifier?

For training framewise phoneme classifier, we align transcript and speech using a forced alignment tool, Montreal Forced Aligner (MFA).
Pretrained MFA models? So implicitly making use of additional data? This should be made very clear. A statement like "Guided-TTS only makes use of LJSpeech for this experiment" is factually incorrect.

"the norm of the unconditional score"
Is this the norm of the gradient or of the log prob, or what exactly?

tts -> TTS. Other English grammar errors (missing articles etc).

HiFi-GAN vocoder, where is it from? pretrained? public?

As I understand, Table 1 uses only LJSpeech (fully transcribed) for all models, including Guided-TTS, except the last row of Guided-TTS, which uses LibriTTS for the phoneme classifier?
Grad-TTS performs better (Table 1).

Table 2 uses only the specified other corpora (e.g. HiFi TTS) for training but not LJSpeech in that case?
Why does it perform worse, even with more data? Because of mismatched conditions?
What would happen when LJSpeech + HiFi-TTS is combined, or other combinations? It should get better overall, also over the LJSpeech-only case, right? This should be verified.

Only those two tables with experiments?
This is way too little. A lot of further things should be analyzed. E.g.:

- What is the effect of the norm-based guidance and the gradient scale s?
- What is the effect of different types of classifiers, e.g. making the classifier stronger or weaker?
- Can we actually improve by the use of more (untranscribed) data? Isn't this the main motivation behind this? That was my understanding. But this is actually not shown, or not even tested. What is the point then in being able to use more data?
- Etc

Where is the code?


**Summary Of The Paper:**

Combines an unconditional denoising diffusion probabilistic model (DDPM) with a separately trained phoneme classifier to be able to guide the generation of mel spectogram towards a given aligned phoneme sequence, thus effectively having a conditional generative model.

This is based on the same principles which were applied before on images (Song et al. (2021b)).

**Summary Of The Review:**

While the novelty is limited, this is still a very interesting direction to investigate. However, in its current form, it provides way too little experiments and analysis.

---

> ### Author Response · Authors · 2021-11-23
> **Response to  Reviewer 6P3J (1/3)**
>
> We would like to thank you for your valuable comments. We've revised the paper according to the reviewers' questions and listed key changes in the general response. Please check the general response.
>
> We admit that it was unclear how Guided-TTS uses untranscribed speech data to build the text-to-speech model in our initial submission. To prevent misunderstanding, we've provided more detailed explanations of Guided-TTS to the captions in Table 1 and Table 2 and improved the readability of the tables.
>
> We will provide detailed explanations about Table 1 and Table 2. Then, we will answer the remaining questions. (novelty, analysis, the role of untranscribed speech data, results, some missing details)
>
> Guided-TTS has an advantage in that it can separate the training of each module, and each module can be trained on different datasets. Table 1 shows both the case where each module of Guided-TTS is trained on the same dataset (Guided-TTS-T) and the case where it is trained on different datasets (Guided-TTS-U). In Table 1, when the phoneme classifier is trained on the same dataset as the unconditional DDPM, transcribed data for LJSpeech dataset is required, so we named it Guided-TTS-T (T: Transcribed). In the case of Guided-TTS-U (U: Untranscribed), unconditional DDPM is trained on the untranscribed LJSpeech dataset, but the rest of the modules are trained on other datasets. The training datasets for each module are listed below:
>
> **Guided-TTS-T**
> - unconditional DDPM: Trained on LJSpeech (w/o transcript)
> - phoneme classifier: Trained on LJSpeech (w/ transcript)
> - duration predictor: Trained on LJSpeech (w/ transcript)
> **Guided-TTS-U (We do not use any transcript of LJSpeech.)**
> - unconditional DDPM: Trained on LJSpeech (w/o transcript)
> - phoneme classifier: Trained on LibriTTS (w/ transcript)
> - duration predictor: Trained on LibriTTS (w/ transcript)
> - speaker encoder: Trained on VoxCeleb 2
>
> As shown above, Guided-TTS-U does not use any transcript for LJSpeech at all. We focused on showing comparable performance to other baseline models using the transcribed data for LJSpeech. In other words, our goal is not to improve performance by further using untranscribed speech data. While existing TTS models cannot be trained without transcribed data for the desired speaker, our work shows that a high-quality TTS system for the target speaker can be achieved without any transcript of that speaker.
>
> In Table 2, we report the results of Guided-TTS-U (U: Untranscribed) for various speakers to show that we can construct a TTS model with untranscribed speech data for more diverse speakers. We trained unconditional DDPM for the LJSpeech, 3 speakers in Hi-Fi TTS, and Blizzard 2013, respectively. Each Guided-TTS-U is a TTS model for the speaker of untranscribed speech data on which each unconditional DDPM is trained. We did not mix datasets from any two speakers. The training dataset for each module in Table 2 is as follow:
>
> **Guided-TTS-U (UNTRANSCRIBED DATA)**
> - UNTRANSCRIBED DATA in [LJSpeech, Hi-Fi TTS Spk ID 92, Hi-Fi TTS Spk ID 6097, Hi-Fi TTS Spk ID 9017, Blizzard 2013]
> - Unconditional DDPM: Trained on UNTRANSCRIBED DATA (w/o transcript)
> - Phoneme classifier: Trained on LibriTTS (w/ transcript)
> - Duration predictor: Trained on LibriTTS (w/ transcript)
> - Speaker encoder: Trained on VoxCeleb 2
>
> All Guided-TTS-U models use the same phoneme classifier, duration predictor, and speaker encoder to guide the unconditional DDPMs trained on various speaker datasets, and through this, we showed the possibility of constructing a TTS model without any transcript for various desired speakers with sufficient untranscribed speech data.

---

> ### Author Response · Authors · 2021-11-23
> **Response to Reviewer 6P3J (2/3)**
>
> ### Novelty
>
> It is true that Guided-TTS extends the conditioning method proposed in [1] to TTS. However, our method differs from [1] in two respects:
>
> **Guided-TTS-U uses different datasets for training unconditional DDPM and other modules.**
>
> In [1], the generative model and the classifier are trained on the same image dataset (CIFAR10) for a conditional generation. If this is naively extended to TTS, it corresponds to Guided-TTS-T (T: Transcribed) and does not have a great advantage in that transcribed data for the desired speaker is required. However, the biggest advantage of our work comes when we train the unconditional DDPM and the phoneme classifier on different datasets. In the case of Guided-TTS-U (U: Untranscribed), we train the phoneme classifier on the large-scale transcribed data, LibriTTS, and we train the unconditional DDPM on the untranscribed data of the desired speaker for TTS. Thus, we can construct a TTS system for a specific speaker without any transcript for that speaker.
>
> **Norm-based guidance**
>
> With the classifier guidance presented in [1], we found that the classifier's guiding effect was small as it was ignored when the gradient norm of the classifier was much smaller than the norm of the unconditional score. It occurs some pronunciation issues in which phoneme sequences were mispronounced in certain parts. Therefore, we propose norm-based guidance that does not ignore the guiding effect of a classifier, and we could solve the problem of mispronunciation of a given phoneme sequence. Please check Section 5.3 for detailed explanations.
>
> ### Analysis
>
> **The effect of the norm-based guidance and the gradient scale s**
>
> The analysis on the effect of norm-based guidance seems much important to show the novelty of our work. We added Section 5.3 about the analysis on the effect of norm-based guidance and the gradient scale s. We measured the Automatic Speech Recognition (ASR) metrics of generated samples to check whether our model synthesizes speech samples according to the given transcript accurately. Please check Section 5.3 for detailed explanations.
>
> **The effect of different types of classifiers**
>
> In Guided-TTS-U, we used a large-sized phoneme classifier to guide the unconditional DDPM for any speaker and to learn well enough on the LibriTTS dataset. Our phoneme classifier is a WaveNet-like structure with 512 residual channels. To show the effect of the classifier according to its size, we trained a small classifier with the residual channel of 256, which we refer to "small phoneme classifier".
>
> We measured the ASR metrics of generated samples for each case. The results for Guided-TTS-U using each classifier are shown in the tables below. The upper table shows the results of Guided-TTS-U in the paper and the lower table shows the results of Guided-TTS-U (w/ small phoneme classifier).  According to ASR metrics, we can see that the size of the classifiers matters to guide unconditional DDPM accurately. As shown in Table 1 in the paper, Guide-TTS-U shows comparable ASR metrics to other TTS models, which indicates that our phoneme classifier is large enough to guide unconditional DDPM well. Thus, we only provide the results of Guided-TTS-U (w/ small phoneme classifier).
>
> <Table S1.> ASR metrics of Guided-TTS-U according to gradient scale s
>
>   - unconditional DDPM trained on LJSpeech (w/o transcript)
>   - phoneme classifier: WaveNet-like structure with 512 residual channels
>
>   |Gradient scale|0|0.1|0.2|0.3|0.4|0.5|0.6|0.7|0.8|0.9|1.0|
>   |-----------|:-----------:|:-----------:|:-----------:|:-----------:|:-----------:|:-----------:|:-----------:|:-----------:|:-----------:|:-----------:|:-----------:|
>   |CER(%)|77.57|10.67|2.55|1.26|1.05|1.35|1.35|1.28|1.50|1.52|4.05|
>   |WER(%)|103.77|22.51|7.54|4.61|4.19|4.40|5.03|4.82|5.03|5.13|8.17|
>
>
>
> <Table S2.> ASR metrics of Guided-TTS-U (w/ small phoneme classifier) according to gradient scale s
>
>   - unconditional DDPM trained on LJSpeech (w/o transcript)
>   - small phoneme classifier: WaveNet-like structure with 256 residual channels
>
>   |Gradient scale|0|0.1|0.2|0.3|0.4|0.5|0.6|0.7|0.8|0.9|1.0|
>   |-----------|:-----------:|:-----------:|:-----------:|:-----------:|:-----------:|:-----------:|:-----------:|:-----------:|:-----------:|:-----------:|:-----------:|
>   |CER(%)|77.35|8.93|3.41|2.21|2.08|1.61|1.72|1.74|1.67|1.82|11.85|
> |WER(%)|103.25|20.21|8.38|6.91|7.02|5.65|5.55|6.07|5.03|5.76|14.87|
>
>
> [1] Song et al., "Score-Based Generative Modeling through Stochastic Differential Equations"

---

> ### Author Response · Authors · 2021-11-23
> **Response to Reviewer 6P3J (3/3)**
>
> ### The role of untranscribed speech data
>
> **Q1)**  As I understand, Table 1 uses only LJSpeech (fully transcribed) for all models, including Guided-TTS, except the last row of Guided-TTS, which uses LibriTTS for the phoneme classifier? Grad-TTS performs better (Table 1)
>
> **Q2)** Can we actually improve by the use of more (untranscribed) data? Isn't this the main motivation behind this? That was my understanding. But this is actually not shown, or not even tested. What is the point then in being able to use more data?
>
> **A)** Our work is not intended to improve the performance of TTS models using more untranscribed speech data. We aim to build a TTS model for the target speaker without transcripts of that speaker by leveraging a large-scale transcribed speech dataset. Note that most previous TTS models cannot be trained without transcribed data for the target speaker.
>
> As explained above, Guided-TTS-U (U: Untranscribed) uses untranscribed speech data of the desired speaker only when we train unconditional DDPM, and the other modules are trained on LibriTTS or VoxCeleb 2, so it does not use any transcript of that speaker.
>
> In Table 1, Guided-TTS-U shows comparable performance to previous TTS models, which require transcribed speech data for the desired speaker. In Table 2, we additionally show that once we train a phoneme classifier on a large-scale transcribed dataset, TTS models for various speakers can be achieved by only training unconditional DDPM with untranscribed speech data for those speakers.
>
> ### The results in Table 2
>
> **Q)** Table 2 uses only the specified other corpora (e.g. HiFi TTS) for training but not LJSpeech in that case? Why does it perform worse, even with more data? Because of mismatched conditions? What would happen when LJSpeech + HiFi-TTS is combined, or other combinations? It should get better overall, also over the LJSpeech-only case, right? This should be verified.
>
> **A)** We admit that there was a lack of explanation for Table 2, and added an explanation for each model. We trained unconditional DDPMs on various untranscribed datasets, and guided the unconditional models using the same phoneme classifier, duration predictor, and speaker encoder. In Table 2, we specified the dataset on which each unconditional DDPM was trained. In order for the unconditioned DDPM to model speech of the target speaker, we trained unconditional DDPM on the untranscribed speech data for the target speaker, and did not mix any untranscribed speech dataset for training.
>
> Also, each model in Table 2 is a TTS model for a different target speaker. For these models, we measured the MOS by generating samples for 50 randomly chosen sentences in the test set of LJSpeech. We did not intend to compare the MOS of each model for different speakers, but to show that we can achieve TTS for each target speaker by only training the unconditional DDPM on untranscribed speech of the corresponding target speaker.
>
> ### Some missing details & explanations
>
> **Q)** HiFi-GAN vocoder, where is it from? pretrained? public?
>
> **A)** We used the pre-trained Hifi-GAN from the Hifi-GAN repository ([1]). We used "LJ_FT_T2_V1" among the pre-trained weights of the Hifi-GAN repository for LJSpeech dataset, and "UNIVERSAL_V1" for Hifi-TTS dataset and Blizzard 2013 dataset.
>
> **Q)** For training framewise phoneme classifier, we align transcript and speech using a forced alignment tool, Montreal Forced Aligner (MFA). Pretrained MFA models? So implicitly making use of additional data? This should be made very clear. A statement like "Guided-TTS only makes use of LJSpeech for this experiment" is factually incorrect.
>
> **A)** We didn't include training details of MFA in the initial submission. We added this content to Appendix A.2. We trained MFA ([2]) to extract the alignment using the same dataset, which is used to train the phoneme classifier and duration predictor. That is, Guided-TTS-T for LJSpeech uses MFA trained on LJSpeech, and Guided-TTS-U for any speaker uses MFA trained on LibriTTS. Thus, Guided-TTS-T in Table 1 only makes use of LJSpeech.
>
> **Q)** Unconditional DDPM can be trained unsupervisedly, yes. But not the phoneme classifier?
>
> **A)** The phoneme classifier is trained on transcribed data. But, we don't have to train the phoneme classifier on the transcribed data of the target speaker. Thus, in Guided-TTS-U, we leverage a large-scale multi-speaker transcribed dataset to train the phoneme classifier.
>
> **Q)** "the norm of the unconditional score" Is this the norm of the gradient or of the log prob, or what exactly?
>
> **A)** The definition of the score is the gradient of log density with respect to the data. "The norm of the unconditional score" is the norm of the gradient of the log density, Norm(\nabla_X_t log p(X_t)).
>
> **Code)** We will release the code and the pre-trained model on GitHub when it is published.
>
> [1] https://github.com/jik876/hifi-gan
>
> [2] https://github.com/MontrealCorpusTools/Montreal-Forced-Aligner

---

> > ### Comment · Reviewer_6P3J · 2021-11-29
> > **Thank you for the feedback**
> >
> > Thank you for the detailed feedback. This clarifies many of my questions.
> >
> > Unfortunately I still don't see this good enough for acceptance. I think it needs some rewriting, maybe even title change, also abstract change, to make the motivation more clear because I totally misunderstood this.
> >
> > Also, I don't think this is too relevant for the ICLR community then. This is too narrow. I think some speech conference would be more appropriate.
> >
> > But my suggestion is, why don't you widen the scope? I would actually analyze what I thought what the motivation is, i.e. to use untranscribed speech to improve the TTS performance in general. In principle, if I did not misunderstood anything, you can very easily test this, by just using more untranscribed data.

---

### Official Review · Reviewer_MW7v · 2021-11-02

**Correctness:** 2
**Technical Novelty And Significance:** 2
**Empirical Novelty And Significance:** 3
**Recommendation:** 5
**Confidence:** 5

**Main Review:**

Strengths:

(1) I really like the direction, doing TTS by decoupling audio modeling, text (phoneme) modeling and speaker modeling. This could potentially  utilize large amount data and respect to each task's data requirement, for example, audio modeling could use much more noisy data.
(2) The way it adding gradients from phoneme classifier to guide unconditional model is new to speech application. The paper also propose practical trick to make it works better as norm based guidance.

Weakness:

(1) The paper doesn't really prove having additional (more diversified dataset) is better. From Table 1, using LibriTTS and LJSpeech looks about the same. To train the phoenme classifier it still need the whole LJSpeech dataset. Table 2 trying to emphasis the model can decouple generative model training and the classifier. However, it doesn't compare with a proper baseline, for example, you can also train a vocoder and mel-predictor plus speaker emb on different dataset.

(2) The paper doesn't answer how stable of such a model. One potential issue of decoupled model is the TTS cannot always follow the given text. It's better to report the ASR metric, so it would make sure the MOS wasn't misleading, for example. it could generate human sound, but with lots of pronunciation error and still get good MOS score.

(3) Following on (1), it's unclear to me for table 2 what data used to train classifier and what data for DPM.

(4) In the related section, it's better to compare with non e2e model, given the fact this model use phoneme classifier, aligner (to get duration) and a vocoder like DPM. As a comparison, traditional tts (before tacotron) model will first train a model to get linguistic features and duration and then use a vocoder on top of it.


Other comments:
(1) wihtout -> without in introduction.
(2) Table 1 and 2 was not very clear to me. Can you explain "In addition, we show that we can generate high quality samples using untranscribed speech of speaker", does it mea your phoneme classifier is trained on different data? It's better to mark it in the table.

**Summary Of The Paper:**

This paper borrows the backbone from Grad-TTS [1], building an unconditional diffusion based model on speech input. With a phone classifier and duration predictor trained on transcribed data, and a speaker encoder trained on labeled speaker dataset, the model was able to synthesize speech on given text. Experimental results shows it's comparable with Grad-TTS [1] trained on transcribed data (by conditioning on the text during training). Ablation shows it can generalize to a diversified dataset.


[1] Vadim Popov, Ivan Vovk, Vladimir Gogoryan, Tasnima Sadekova, and Mikhail Kudinov. Grad-TTS:
A Diffusion Probabilistic Model for Text-to-Speech. In Proceedings of the 38th International
Conference on Machine Learning, ICML 2021, 18-24 July 2021, Virtual Event, volume 139 of
Proceedings of Machine Learning Research, pp. 8599–8608. PMLR, 2021a.

**Summary Of The Review:**

The biggest advantage of the proposed framework is it was able to use large amount speech data so it potentially could modeling wider range of prosody or other aspect of speech. However, the paper's experimental results cannot well-support it. I would happy to change the score if the author can improve follows:

(1) Adding evidence such guidance framework is relative stable, e.g. faithfully reflect the text it been given.
(2) Demonstrate this model is better than a simpler baseline: a text-to-mel model + speaker verification network + vocoder (trained on diversified mel). My bet is if here is a win, DPM model might capture better prosody. If showing prosody win is hard, at least demonstrate this model give better results on Blizzard which i assume the training data is more noisy?
(3) The readability of Table 1 and 2 can be improved. E.g. adding what data been trained for the classifier, which speaker been used etc.

---

> ### Author Response · Authors · 2021-11-23
> **Response to Reviewer MW7v**
>
> We would like to thank you for your valuable comments. We've revised the paper according to the reviewers' questions and listed key changes in the general response. Please check the general response.
>
> ### Readability of Table 1, 2
>
> We've provided more detailed explanations of Guided-TTS to the captions in Table 1 and Table 2. We also added whether each model uses the transcript of the target speaker (LJSpeech) in Table 1, and specified the untranscribed datasets used to train unconditional generative models of Guided-TTS in Table 2.
>
> In Table 1, we compared Guided-TTS and baselines for the LJSpeech dataset. In Guided-TTS, unconditional DDPM and other modules can be trained on the same dataset or on different datasets. To prevent misunderstanding, Guided-TTS-T (T: Transcribed) is specified for the case of training the unconditional DDPM and other modules on the same data. That is, Guided-TTS-T requires transcribed data of LJSpeech to train the phoneme classifier and the duration predictor. In the case of Guided-TTS-U (U: Untranscribed), we trained unconditional DDPM on LJSpeech, and the phoneme classifier and the duration predictor are trained on the large-scale transcribed speech data, LibriTTS. Thus, we do not use any transcript of LJSpeech for training each module of Guided-TTS-U.
>
> Guided-TTS-T achieves comparable performances to existing TTS models, which shows that TTS via the classifier guidance works well and it can be a new option to build TTS models. Through the performance of Guided-TTS-U, we demonstrated that our proposed model can achieve comparable performance with existing TTS models without any transcript of LJSpeech.
>
> In Table 2, we specified the training dataset of each module and added detailed captions for readability.
>
> ### Stability of guidance
>
> The contents related to the stability of guidance seem much important to show the novelty of the proposed norm-based guidance in our work. We added Section 5.3 related to the stability of guidance and reported the ASR metrics as well as MOS measured by the pre-trained ASR model in Table 1. The ASR metrics are measured by the pre-trained CTC-based conformer model available in [1].
>
> In the early stages of our work, we observed that Guided-TTS with the previous guidance method in [2] produces mispronounced samples given text. For the stability of the guidance, we proposed the norm-based guidance and provided the analysis on the effect of norm-based guidance in Section 5.3. The result of Guided-TTS using the norm-based guidance is shown in Table 1. The ASR metrics shown in Table 1 are as below:
>
> <Table S1.> ASR metrics (CER/WER) of each method
>
>   |Method|CER(%)|WER(%)|
>   |-----------|:-----------:|:-----------:|
>   |GT wav|0.79|3.77|
>   |GT mel + Hifi-GAN|1.05|4.08|
>   |Glow-TTS|1.09|5.03|
>   |Grad-TTS|1.31|5.55|
>   |Guided-TTS-T|1.20|4.71|
>   |Guided-TTS-U|1.26|4.61|
>
>
> As shown in the table above, Guided-TTS with the norm-based guidance shows similar ASR metrics compared to the existing TTS models, which indicates that Guided-TTS synthesizes the samples according to transcript accurately. Additionally, we also compared the proposed norm-based classifier guidance with the classifier guidance used in [2]. The results of both guidance methods are shown in Fig. 2 in Section 5.3. Please check Section 5.3 for detailed explanations.
>
> ### Comparison with the previous TTS model using speaker verification models
>
> At the initial submission, we compared Guided-TTS with the stronger baselines (well-performed TTS models) rather than with few-shot voice cloning models using untranscribed speech data for adaptation. We aimed to achieve comparable performance with the stronger baselines by directly modeling the characteristics of the target speaker rather than simply modeling the timbre of the target speaker.
>
> For comparison, we choose the pre-trained model provided by [3] (# of Github stars > 30K) as a baseline for few-shot TTS, which implemented the SV2TTS model proposed in [4]. This pre-trained model uses a 5-second speech of the target speaker to extract the speaker embedding for a few-shot TTS. Since the repository uses the pre-trained WaveRNN, not Hifi-GAN, a direct comparison of "sample quality" is meaningless, and it would be good to focus on the characteristics of the speaker. (prosody & American/British accent)
>
> We uploaded the samples of Guided-TTS and the baseline on the demo page. While the baseline cannot capture the British accent of the female speaker in Hi-Fi TTS (spk id: 92), Guided-TTS can model the British accent of the speaker. Please listen to the samples on the [demo page](https://bit.ly/3oWhVJg).
>
> [1] https://github.com/NVIDIA/NeMo
>
> [2] Song et al., "Score-Based Generative Modeling through Stochastic Differential Equations"
>
> [3] https://github.com/CorentinJ/Real-Time-Voice-Cloning?ref=pythonrepo.com
>
> [4] Jia et al., "Transfer Learning from Speaker Verification to Multispeaker Text-to-Speech Synthesis"

---

### Official Review · Reviewer_XRxC · 2021-11-03

**Correctness:** 3
**Technical Novelty And Significance:** 3
**Empirical Novelty And Significance:** 3
**Recommendation:** 5
**Confidence:** 4

**Main Review:**

The proposed Guided-TTS model utilizes both diffusion probabilistic model with the gradient signals from phoneme classifier, duration predictor and speaker encoder. The overall architecture is more like pretraining + finetuning or multitask learning (if jointly trained), which is not exactly 'generate speech from untranscribed speech data', since both untranscribed and transcribed data are used in the process.

Besides that, one of the main benefits of using untranscribed speech data, which is to reduce the amount of annotated transcribed data required, is not showed in the paper. In the experiment results, the pattern of data annotation reduction is not illustrated and discussed.

**Summary Of The Paper:**

This paper presents Guided-TTS which is a TTS model that learns to generate speech from untranscribed speech data. Guided-TTS combines an unconditional diffusion probabilistic model with a separately trained phoneme classifier to guide the generative
process of mel-spectrograms from the conditional distribution given transcript.

**Summary Of The Review:**

Overall, this paper proposes a new architecture of utilizing both untranscribed and transcribed speech data to generate speech with good quality.

---

> ### Author Response · Authors · 2021-11-23
> **Response to Reviewer XRxC**
>
> We would like to thank you for your valuable comments. We've revised the paper according to the reviewers' questions and listed key changes in the general response. Please check the general response.
>
> ### Regarding the expression "Generate speech from untranscribed speech data"
>
> Our generative model learns to model the unconditional distribution of speech p(speech) using untranscribed speech data. That's why we used the expression "learns to generate speech from untranscribed speech data". We use transcribed data to train the phoneme classifier and the duration predictor, and these modules guide our unconditional generative model to "generate speech given transcript" only during inference.
>
> We admit that it was unclear how Guided-TTS uses untranscribed speech data to build the text-to-speech model in our initial submission. To prevent misunderstanding, we've provided more detailed explanations of Guided-TTS to the captions in Table 1 and Table 2 and improved the readability of the tables.
>
> As you said, our method requires untranscribed data and transcribed data to train unconditional DDPM and phoneme classifier, respectively. But, our goal is not to build a text-to-speech model without transcribed data at all. We aim to "construct a text-to-speech for the target speaker with untranscribed speech data of the target speaker".
> In Guided-TTS, we can train the unconditional DDPM and the other modules on the same dataset or on different datasets. Depending on whether the transcripts of the target speaker for TTS are used or not, we named our models Guided-TTS-T (T: Transcribed) and Guided-TTS-U (U: Untranscribed), respectively. Guided-TTS-T is specified for the case of training the unconditional DDPM and other modules on the same data. That is, Guided-TTS-T requires transcribed data of LJSpeech to train the phoneme classifier and the duration predictor. In the case of Guided-TTS-U, we trained unconditional DDPM on LJSpeech, and the phoneme classifier and the duration predictor are trained on the large-scale transcribed speech data, LibriTTS. Thus, we do not use any transcript of LJSpeech for training each module of Guided-TTS-U.
> In Table 2, we reported the results of Guided-TTS-U for various speakers (LJSpeech, Hi-fi TTS, Blizzard 2013) to show that we can construct a TTS model with untranscribed speech data for diverse target speakers. Thus, we expressed our goal to build a text-to-speech model using "untranscribed speech data for the desired speaker".
> In addition, the approach is different from finetuning after large-scale pretraining using a large amount of unlabeled data, such as GPT in NLP or wav2vec for speech representation learning. Our method uses the same amount of untranscribed speech data (24 hours for LJSpeech) as that used by the existing TTS models to show that it is possible to build TTS model without any transcript for the desired speaker. We do not finetune unconditional DDPM using text information.
>
> ### Reduction of transcribed data
>
> As mentioned above, Guided-TTS differs from the method of performing text-to-speech by finetuning with a small amount of transcribed speech data after pretraining such as GPT or wav2vec. Guided-TTS-U utilizes the same amount of untranscribed speech data (24 hours for LJSpeech) as that used by the existing TTS models and leverages a large-scale multi-speaker transcribed dataset for classifier guidance. We would like to note that the main contribution of Guided-TTS is to build a TTS model for the target speaker without the transcribed speech data for the target speaker.

---

### Official Review · Reviewer_T6xj · 2021-11-03

**Correctness:** 3
**Technical Novelty And Significance:** 3
**Empirical Novelty And Significance:** 3
**Recommendation:** 8
**Confidence:** 4

**Main Review:**

Strengths of the Paper:
- Proposes a solution to one of the biggest bottlenecks in TTS.
- The idea of using feedback from a separate model as conditioning signal during sampling has been done for image generation, however the application of a phoneme classifier for this in the speech domain is a very elegant solution.
- Experiments of transferring the knowledge of the phoneme-guidance module to an unseen corpus show promising results for training on speech completely without transcription.
- In general, convincing experiments and results

Weaknesses of the Paper:
- As stated in section 5.1, the model needs to be quite large. This probably also means expensive computation during both training and inference. There are no mentions of training time, training hardware or real-time-factor and hardware during inference. The latter of which is quite important for TTS.
- The phoneme recognizer still needs paired data in order to be trained. And while it seems to work well cross-speaker and even cross-accent and cross-gender as the demo page shows, I assume it would work much less well across e.g. languages or highly expressive domains. So paired data would still be needed for those applications, which are the ones that suffer the most from low-resource.

Comments:
- The weakness of paired data being required for the phoneme classifier in challenging domains could potentially be fixed by auto-alignment frameworks, that learn alignments self- or semi-supervised. Those tend to deliver less accurate results however.  Are the accurate alignments of an aligner such as the MFA used in the paper required? I would be interested in the quality-drop-off given less accurate alignments for training the phoneme recognizer, since the guided sampling seems quite complex. In other words, how precise does the guidance of the phoneme classifier have to be?
- I find the level of control that the use of a duration predictor offers intriguing. I think it would be interesting to see how well the unconditional model can handle unnatural conditions, such as holding the same phoneme for multiple seconds.

**Summary Of The Paper:**

The authors propose a denoising diffusion probabilistic model (DDPM) to learn to produce natural spectrograms from noise without a condition. This enables them to train a generative model on unlabelled speech data. They show the effectiveness of their approach by inpainting masked out parts of a spectrogram and by showing off audio samples of unconditioned spectrogram babble, that has been vocoded to a waveform. They further propose a phoneme classification module that serves as conditioning signal for the DDPM during sampling in order to generate spectrograms that match a given phoneme sequence, turning the unconditional DDPM into a text-to-speech model, which the authors call Guided-TTS. This allows for all components of the model to be trained individually on different datasets, alleviating the need for large labelled datasets for TTS.

**Summary Of The Review:**

This paper presents a novel method for TTS with convincing experiments and results. Although it has some weaknesses, I tend to see it marginally above the acceptance threshold.

---

> ### Author Response · Authors · 2021-11-23
> **Response to Reviewer T6xj**
>
> We would like to thank you for your valuable comments. We've revised the paper according to the reviewers' questions and listed key changes in the general response. Please check the general response.
>
> ### Computing resources and real-time factor
>
> Guided-TTS has a total of 4 modules: unconditional DDPM, phoneme classifier, duration predictor, and speaker encoder. We'll explain the training cost of each module. Note that each module is trained separately from the other so that it can be trained on different GPUs at the same time. Also, we would like to note that we train the phoneme classifier (LibriTTS), the duration predictor (LibriTTS), and the speaker encoder (VoxCeleb2) only once to guide the unconditional DDPM for any speaker. We will publish our official code and pre-trained weights for each model when published.
>
> The GPU we used for time measurement was based on NVIDIA's Quadro RTX 8000 GPU with 48GB of VRAM. The training cost and time of each module are shown in Table S1. To help the comparison with previous TTS models, we also report the training cost of Glow-TTS and Grad-TTS.
>
> <Table S1.> Training costs of each module
>
> | Model        |       Module       |  Dataset  | Batch Size | Iterations | Required VRAM | Precision |    Training Time   |
> |--------------|:------------------:|:---------:|:----------:|:----------:|:-------------:|:---------:|:------------------:|
> | Guided-TTS-T | Unconditional DDPM |  LJSpeech |     16     |    780K    |     10.5GB    |    fp16   |      3.5 days      |
> |              | Phoneme Classifier |  LJSpeech |     64     |    195K    |     7.8GB     |    fp16   |     18.3 hours     |
> |              | Duration Predictor |  LJSpeech |     64     |    3.9K    |     3.6GB     |    fp16   |     8.7 minutes    |
> | Guided-TTS-U | Unconditional DDPM |  LJSpeech |     16     |    780K    |     10.5GB    |    fp16   |      3.5 days      |
> |              | Phoneme Classifier |  LibriTTS |     64     |    780K    |  28GB x 4GPUs |    fp16   | 3.75 days x 4GPUs  |
> |              | Duration Predictor |  LibriTTS |     64     |   110.8K   |     9.2GB     |    fp16   |     8.84 hours     |
> |              |   Speaker Encoder  | VoxCeleb2 |     80     |    300K    |     < 2GB     |    fp32   |      0.6 days      |
> | Glow-TTS     |          -         |  LJSpeech |     16     |    240K    |      14GB     |    fp16   |      3.9 days      |
> | Grad-TTS     |          -         |  LJSpeech |     64     |    1.7M    |     < 8GB     |    fp32   |      5.9 days      |
>
> Since the paper focuses on enabling TTS using untranscribed speech for the desired speaker, we didn't report the real-time factor (RTF) in the initial submission. As you mentioned, it would be better to report RTF for better understanding. RTF was calculated based on 50 iterations that we used for our evaluation. and the result is as follows:
>
> <Table S2.> Real-time factor for each component
>
> | Model        | Component                                  |  RTF |
> |--------------|--------------------------------------------|:----:|
> | Guided-TTS-T | total                                      | 0.56 |
> |              | $\nabla_{X_t}\log{p_{\theta}(X_t)}$        | 0.39 |
> |              | $\nabla_{X_t}\log{p_{\phi}(\hat{y}\|X_t)}$ | 0.17 |
> | Guided-TTS-U | total                                      | 0.62 |
> |              | $\nabla_{X_t}\log{p_{\theta}(X_t)}$        | 0.39 |
> |              | $\nabla_{X_t}\log{p_{\phi}(\hat{y}\|X_t)}$ | 0.23 |
>
> Although fast inference speed is not the main target of our paper, Guided-TTS can generate high-quality speech faster than in real-time. We added this content to Appendix A.3. For faster inference, it would be an interesting direction to apply the method to Guided-TTS, which reduce the number of reverse steps of DDPM ([1], [2], [3])
>
> ### Phoneme classifier in challenging domains
>
> In this work, we focus on the case in which a paired dataset exists enough to extract the alignment accurately. As you said, it is an important issue when it is difficult to obtain accurate alignment due to few paired datasets (e.g., low-resource language). As future work, we will proceed with Guided-TTS with high performance even using a paired dataset with a small amount of data. Thanks for the valuable comment.
>
> ### Control the duration and handle the unnatural conditions
>
> As Guided-TTS uses the duration predictor like other non-autoregressive TTS models, we can control the length in the same way as other models. We added samples generated by adjusting duration by [0.8, 0.9, 1.0, 1.1, 1.2] times, and samples with a long duration for a specific phoneme to show how our model handles unnatural conditions in the sentence on the [demo page](https://bit.ly/3oWhVJg).
>
> [1] Watson et al., “Learning to efficiently sample from diffusion probabilistic models”
>
> [2] Kong et al., “On fast sampling of diffusion probabilistic models”
>
> [3] Jolicoeur-Martineau et al., “Gotta go fast when generating data with score-based models”

---

### Author Response · Authors · 2021-11-23
**General Response to All Reviewers**

Thanks to all reviewers for your valuable comments on our work. We revised the writing of the initial submission for a better explanation. The paper has been revised as follows:

### 1. Revise Table 1 and Table 2 (All reviewers)

- We improved the explanations in Section 5.1 and 5.2 to better understand our results presented in Table 1 and Table 2.
- Depending on whether the transcripts of the target speaker for TTS are used or not, we named our models Guided-TTS-T (T: Transcribed) and Guided-TTS-U (U: Untranscribed), respectively.
- In Table 1, we added the ASR metrics of each model to show that Guided-TTS generates samples given text accurately with the proposed guidance method.
- In Table 2, we specified untranscribed datasets for training unconditional generative models of Guided-TTS-U. The phoneme classifier and the duration predictor are trained on the LibriTTS dataset and the speaker encoder is trained on the VoxCeleb 2 dataset.

### 2. Add subsection 5.3 "Analysis on norm-based guidance" (Reviewers MW7v and 6P3J)

- Reviewers MW7v and 6P3J asked questions about the analysis of the stability and the effect of the proposed norm-based guidance. The content related to the questions seems more important to understand our proposed model compared to the inpainting results. Thus, we added the corresponding results in Section 5.3.
- To show the effect of the norm-based guidance, we added the result of the ASR metric for the generated samples from Guided-TTS.
- The inpainting results have been moved to Appendix A.1.

### 3. More samples on the [demo page](https://bit.ly/3oWhVJg)

- Samples from Guided-TTS-U and the publicly available few-shot TTS model ([1]). (Reviewer MW7v)
- Samples for Guided-TTS using the previous classifier guidance ([2]) and the proposed norm-based guidance (Related to Section 5.3)
- Samples generated by controlling the output of the duration predictor (Reviewer T6xj)

### 4. Real-time factor of Guided-TTS in Appendix A.3 (Reviewer T6xj)

### 5. Plot of the norm of the unconditional score and the gradient norm of the classifier according to timestep $t$ in Appendix A.4 (Related to Section 3.2.1)

We will give more detailed answers to each reviewer.

[1] https://github.com/CorentinJ/Real-Time-Voice-Cloning

[2] Song et al., "Score-Based Generative Modeling through Stochastic Differential Equations"

---

### Public Comment · ~Jisoo_Lee1 · 2022-02-14
**Question about evaluation metric**

Dear authors,

Thank you for the nice work. I am trying to reproduce your paper and just have a question for better clarification.

The paper mentioned, “We use the checkpoint of the epoch with the best metric for the phoneme classifier”. Could you add more details on the metric for the phoneme classifier? I guess, phoneme accuracies for all the step t’s should be considered. Am I right?

Thanks.

---

### Decision · Program_Chairs · 2022-01-20

**Decision:**

Reject

**Comment:**

The paper proposes using unlabelled speech data for TTS by decoupling parts of the model.
However, all reveiwers agree that the technique is already known and the experimental results are not strong enough to make advantage of training on more data.
A reject.